# Hypermethylation Loci of ZNF671, IRF8, and OTX1 as Potential Urine-Based Predictive Biomarkers for Bladder Cancer

**DOI:** 10.3390/diagnostics14050468

**Published:** 2024-02-21

**Authors:** Yuan-Hong Jiang, Yu-Shu Liu, Yu-Chung Wei, Jia-Fong Jhang, Hann-Chorng Kuo, Hsin-Hui Huang, Michael W. Y. Chan, Guan-Ling Lin, Wen-Chi Cheng, Shu-Chuan Lin, Hung-Jung Wang

**Affiliations:** 1Department of Urology, Hualien Tzu Chi Hospital, Tzu Chi University, Hualien 970374, Taiwan; redeemerhd@gmail.com (Y.-H.J.); alur1984@hotmail.com (J.-F.J.); hck@tzuchi.com.tw (H.-C.K.); 2Guzip Biomarkers Corporation, Hsinchu City 302041, Taiwan; yushuliu@phalanxbiotech.com (Y.-S.L.); hsinhuih@phalanxbiotech.com (H.-H.H.); pollylin@phalanxbiotech.com (S.-C.L.); 3Phalanx Biotech, Hsinchu City 302041, Taiwan; 4Graduate Institute of Statistics and Information Science, National Changhua University of Education, Changhua City 500207, Taiwan; weiyuchung@cc.ncue.edu.tw; 5Department of Biomedical Sciences, National Chung Cheng University, Minhsiung, Chiayi 621301, Taiwan; biowyc@ccu.edu.tw (M.W.Y.C.); guan0223@gmail.com (G.-L.L.); 6Epigenomics and Human Disease Research Center, National Chung Cheng University, Minhsiung, Chiayi 621301, Taiwan; 7Center for Innovative Research on Aging Society (CIRAS), National Chung Cheng University, Min-Hsiung, Chiayi 621301, Taiwan; 8Institute of Medical Sciences, Tzu Chi University, Hualien 970374, Taiwan; wccheng888@gmail.com; 9Doctoral Degree Program in Translational Medicine, Tzu Chi University and Academia Sinica, Hualien 97004, Taiwan; 10Department of Biomedical Sciences and Engineering, Tzu Chi University, Hualien 970374, Taiwan

**Keywords:** bladder cancer, DNA methylation, ZNF671, OTX1, IRF8, urine biomarkers, machine learning, liquid biopsy, diagnostic panel, qMSP

## Abstract

Bladder cancer (BCa) is a significant health issue and poses a healthcare burden on patients, highlighting the importance of an effective detection method. Here, we developed a urine DNA methylation diagnostic panel for distinguishing between BCa and non-BCa. In the discovery stage, an analysis of the TCGA database was conducted to identify BCa-specific DNA hypermethylation markers. In the validation phase, DNA methylation levels of urine samples were measured with real-time quantitative methylation-specific PCR (qMSP). Comparative analysis of the methylation levels between BCa and non-BCa, along with the receiver operating characteristic (ROC) analyses with machine learning algorithms (logistic regression and decision tree methods) were conducted to develop practical diagnostic panels. The performance evaluation of the panel shows that the individual biomarkers of ZNF671, OTX1, and IRF8 achieved AUCs of 0.86, 0.82, and 0.81, respectively, while the combined yielded an AUC of 0.91. The diagnostic panel using the decision tree algorithm attained an accuracy, sensitivity, and specificity of 82.6%, 75.0%, and 90.9%, respectively. Our results show that the urine-based DNA methylation diagnostic panel provides a sensitive and specific method for detecting and stratifying BCa, showing promise as a standard test that could enhance the diagnosis and prognosis of BCa in clinical settings.

## 1. Introduction

Bladder cancer (BCa) is a global health issue, ranking as the 10th most prevalent form of cancer worldwide [1,2]. In 2022, over half a million people were diagnosed with new cases and there were over 200,000 deaths worldwide [2]. In the United States alone, over 81,000 adults are expected to be diagnosed with bladder cancer [2].

BCa is a complex disease. The etiology and progression of BCa are associated with multiple risk factors, including genetic factors like age and gender, environmental factors, such as cigarette smoking and chemical exposure, and the interaction between the host and the microbial communities that inhabit the bladder called bladder microbiota [3,4,5]. According to the National Cancer Institute (NCI), the risk of BCa increases with age, and males are more likely to develop BCa than females. Tobacco use, especially cigarette smoking, greatly elevates the risk of developing BCa by exposing the bladder to harmful substances that can lead to DNA damage in its lining cells [6,7]. Additional exposure to certain industrial chemicals, like aromatic amines and arsenic in drinking water, has been associated with a higher risk of BCa [8,9]. The third and emerging risk factor for BCa is the diverse urinary microbiota [10]. The urinary microbiota is the collection of microorganisms that inhabit the urinary tract and may influence the development and progression of BCa. Several studies have reported that changes in the urinary microbiota of BC patients compared to healthy controls, indicating a potential role of these microbes in carcinogenesis, diagnosis, prognosis, and treatment of BCa [10]. For example, Zeng et al. found that BCa patients had a higher richness and diversity of urinary microbiota than healthy individuals, and these differences were linked to the recurrence and stage of BCa. [11]. Moreover, some bacterial taxa, such as *Actinobacteria*, were found to be more abundant in non-tumor bladder mucosa than in tumor tissues, indicating a potential protective effect [12]. Conversely, other bacteria, such as *Fusobacterium* and *Porphyromonas*, were more prevalent in BCa than in health controls, implying a possible pro-tumorigenic role [10,13]. BCa is frequently asymptomatic in its early stages, although it can sometimes be accompanied by painless macroscopic hematuria [14,15,16,17]. BCa is typically diagnosed as non-muscle invasive bladder cancer (NMIBC) in about 75% of patients [18,19]. The 5-year survival rate of NMIBC can reach as high as 92%; however, if NMIBC progresses to muscle-invasive metastatic BCa (MIBC), the survival rate drops significantly to below 45%, making it a leading cause of death and poor outcomes in BCa patients [18]. To diagnose BCa, two common methods used are urine cytology and invasive cystoscopy. Urine cytology is currently used for the detection of BCa and is the gold standard for detecting high-grade tumors and carcinoma in situ. Urine cytology has a high specificity of 90–100%, but it is limited by its low sensitivity of 10–40% for identifying low-grade tumors, which account for 70–80% of all bladder cancers [20,21]. Nevertheless, this diagnostic procedure relies on the expertise of the operator and is affected by various factors including sample quality, inflammation, infection, and instrumentation. Cystoscopy is the main tool used for visualizing bladder lesions and collecting biopsies. This method is largely limited by its invasiveness. It may cause patient discomfort, pain, bleeding, infection, and anxiety. Unfortunately, it might not detect flat lesions or small papillary tumors [22]. Additionally, cystoscopy is expensive and demands both medical facilities and skilled personnel. Given the high recurrence rate of NMIBC, patients are advised to undergo lifelong surveillance through frequent monitoring using cystoscopy, which presents challenges related to healthcare costs and disease management. Therefore, there is an urgent need to develop new biomarkers that can be detected early in a non-invasive way, referred to as a liquid biopsy. The emerging non-invasive liquid biopsy has recently gained attention in diagnosing and monitoring disease progression. In the case of BCa, urine-based biomarkers show promise as valuable tools for cost-effective and non-invasive strategies to detect BCa [23].

Several urine-based biomarkers have been developed and evaluated for bladder cancer diagnosis, ranging from single molecules to multiplex assays based on mRNA, DNA, or non-coding RNA [24]. One of them, microRNAs (miRNAs) are small non-coding RNAs that regulate gene expression and are involved in various biological processes, including cancer development and progression. miRNAs are also stable and abundant in biological fluids, such as urine, where they can be derived from exosomes, cell-free supernatant, or cell pellets. Several studies have shown that miRNAs can act as diagnostic biomarkers for BC in urine and have been extensively reviewed elsewhere [25]. As the above-mentioned, the urinary microbiota or termed urobiome has been considered a new avenue in discovering novel microbiota-derived biomarkers for BCa. Several bacteria specific to BCa have been identified and proven to be valuable biomarkers for BCa detection [4]. In particular, analysis of the urobiome in first-morning (FM) urine samples revealed that *Porphyromonas somerae* serves as a predictive biomarker for BCa patients undergoing transurethral resection of bladder tumor (TURBT) [13].

DNA methylation alternations are considered a cancer hallmark [26]. In bladder cancer, hypermethylation is frequently observed in the promoter regions of numerous cancer-driver genes and plays a critical role in the early stages of tumorigenesis [27]. Thus, in view of the development of a urine-based DNA methylation assay, the assessment of tumor-associated aberrant DNA methylation can be achieved by capturing urine DNAs released from lysed tumor cells or the intact cells exfoliated from bladder tumors [28,29]. Moreover, urine DNA methylation has been demonstrated to exhibit steady and reliable epigenetic indicators when compared to other biomolecules like RNAs, miRNAs, or proteins [30,31,32]. This makes them suitable for the development of a urine-based assay aimed at detecting early-stage BCa. Therefore, several genes and gene sets have been examined to compare DNA methylation levels between BCa tumors, normal controls, and their urine sediments in order to identify potential methylation biomarkers [33,34,35]. Several DNA methylation markers, such as in the genes of OTX1, SOX1-OT, DMRTA2, ONECUT2, and TWIST1, have been identified and characterized in previous studies [35,36,37].

In this study, we initially utilized the TCGA bladder cancer cohort to investigate potential DNA methylation markers associated with BCa in the ZNF671, OTX1, and IRF8 genes. This cohort provides comprehensive information including DNA methylation profiling from methylation microarray and the clinicopathological phenotypes from BCa tumors and normal controls [38], allowing for the identification of the BCa-specific hypermethylated regions. In clinical validation, urine samples were collected from both BCa patients and non-BCa controls to measure the urinary DNA methylation levels. As expected, all three genes showed higher methylation levels in BCa urine compared to non-BCa samples. To construct a predicting model, machine learning algorithms, including logistic regression and decision tree, were employed to develop a non-invasive urine-based DNA methylation diagnostic panel. This diagnostic panel with the three hypermethylated genes provided promising characteristics for early detection of BCa through urine-based DNA methylation analysis.

## 2. Materials and Methods

### 2.1. Study Design

A flow chart depicting the development stages of the diagnostic panel in this study is shown below (Figure 1).

### 2.2. TCGA Dataset Analysis

In the discovery phase, we analyzed the TCGA BLCA (bladder cancer) cohort to identify BCa-associated hypermethylated sites on ZNF671, IRF8, and OTX1 genes [38]. In this cohort, we retrieved and analyzed the data of Illumina Human Methylation 450 K array and clinicopathologic variables from 411 BCa tissue and 17 normal tissue samples through the UCSC Xena website (https://xena.ucsc.edu/) (accessed on 10 May 2022) [39]. The retrieved data were replotted with GraphPad Prism 10 (GraphPad Software, San Diego, CA, USA). Survival curves (Kaplan–Meier plots) for each gene were generated by Kaplan–Meier Plotter server (https://kmplot.com/analysis/) (accessed on 4 February 2024) [40].

### 2.3. Participants and Urine Sample Collection

The study was conducted according to the guidelines of the Declaration of Helsinki and approved by the Institutional Review Board and Ethics Committee of Buddhist Tzu Chi General Hospital, Hualien City, Taiwan (Identifier Number: IRB109-201B). All participants were enrolled in our case–control retrospective study from November 2020 to November 2021. Written informed consent was obtained for all participants. These patients had voided urine collection performed because of gross hematuria followed by cystoscopy or surgery at the Hualien Tzu Chi General Hospital. The demographics and clinicopathologic characteristics of the participants are summarized in Table 1. The voided urine (approximately 50 mL) was collected before cystoscopy or surgery. The collected voided urine samples were immediately placed in a Urine Conditioning Buffer (Zymo Research, Irvine, CA, USA) and stored at −20 °C for future analysis.

### 2.4. Urine DNA Extraction and Bisulfite Conversion

The voided urine (approximately 50 mL) was collected before cystoscopy or surgery. The collected urine samples were mixed with 20 µL of clearing beads (Zymo Research, Irvine, CA, USA) and vortexed thoroughly. Then, the urine mixture samples were centrifuged at 3000× *g* for 10 min and the pellets were extracted DNA by using a Quick-DNA Urine Kit (Zymo Research, Irvine, CA, USA) according to the manufacturer’s instructions. Briefly, the pellet was further treated with pellet digestion buffer containing proteinase K with a continuous vortex at 55 °C for 30 min to digest urinary proteins. To elute the bound urinary DNA, the beads first reacted with genomic lysis buffer, and then the mixtures were transferred into a Zymo-spin IC-S column in a collection tube provided by the kit. After the washing steps, the urinary DNA was then eluted with 20 μL of urine DNA elution buffer. The concentration of the eluted DNA was estimated with Nanodrop Spectrophotometry (Thermo Fisher, Fair Lawn, NJ, USA). The extracted urine DNA (200 ng) was bisulfite-treated by using EZ DNA Methylation-Gold Kit (Zymo Research, Irvine, CA, USA) according to the manufacturer’s instructions. Briefly, the eluted urine DNA was treated with Zymo CT conversion reagent by following these reaction steps: 98 °C for 10 min to denature the DNA, followed by 64 °C for 2.5 h to sulfate unmethylated GC-rich regions, and then stored at 4 °C for the next elution step. The DNA bound to Zymo-Spin IC spin column was further desulphonated and washed on columns with a Zymo IC spin column provided by the kit. The bisulfite-converted DNA was eluted with 20 μL and the DNA was ready for qMSP analysis. The 100% CpG Methylated Human Genomic DNA (Thermo Fisher, Waltham, MA, USA) was used as a positive control for methylation, and nuclease-free water was used as a negative control for bisulfite conversion and downstream qMSP analysis.

### 2.5. Real-Time Quantitative Methylation-Specific PCR (qMSP)

The sequences of primers specific to methylated and non-methylated CpG sites of ZNF671, OTX1, and IRF8 genes were designed through the UCSC browser database and Applied Biosystems Methyl Primer Express Software v1.0 (Applied Biosystems, Waltham, MA, USA) and listed in Appendix A. The methylation status of the bisulfite-converted DNA was measured by qMSP with LightCycler 480 System (Roche, Penzberg, Germany). A non-CpG region of the type II collagen gene (*COL2A1*) was as an internal control to normalize the input of bisulfited DNA. PCR reactions were carried out in 20 µL containing 2 µL bisulfite-converted DNA, 0.1 µM of each primer, and 10 µL master mix using the following thermal profiles: 95 °C for 10 min, 45 cycles of 95 °C for 30 s, 60 °C for 30 s, and 72 °C for 30 s, and cooling at 40 °C for 45 s. For each qPCR run, positive and negative controls were executed, and the Ct values were calculated. The qMSP was performed once for each sample. The *COL2A1* gene, a CpG island-free gene whose copy number was not affected by methylation status in the qMSP assay, was used as an input control [41]. The reaction was considered invalid if the Ct value of *COL2A1* was >32. DNA methylation level of the ZNF671, OTX1, and IRF8 genes was estimated using crossing point (ΔCt) values calculated by the formula: Ct value of candidate gene − Ct value of COL2A1.

### 2.6. Construction of Diagnostic Models with Machine Learning Algorithms

The diagnostic panels for predicting BCa utilizing urine DNA methylation measurements were created by using a dual approach, encompassing traditional statistical logistic regression models and the classification and regression tree (CART) version of decision tree algorithms from the domain of machine learning [42,43]. The decision tree model was specifically chosen from a range of machine learning algorithms for its two capabilities: not only does it predict, but its tree structure also provides a clear understanding of the rules governing BCa prediction based on urine DNA methylation. This clarity offers a significant advantage over the more opaque “black box” machine learning algorithms, facilitating practical understanding and usage in clinical settings.

Initially, logistic regression was used to construct models for each individual methylation marker, examining their significance in differentiating between BCa and non-BCa cases. Furthermore, the three methylation markers were integrated into both logistic regression and decision tree models to investigate their collective predictive power for BCa and non-BCa. Within the decision tree model, nodes were split using a standard method that selected urine DNA methylation and value thresholds to maximize Gini impurity reduction. Pruning techniques were also employed to avoid overfitting in the decision tree model, thereby removing branches with minimal predictive contribution. For model validation, all samples were randomly allocated into training and testing sets, according to the commonly used split ratio of 80%:20%. Five standard metrics were utilized to assess the performance of the constructed models, including accuracy, sensitivity, specificity, precision, and F1 score, applied to both training and testing sets. The training set additionally provided the AUC metric for evaluation. The entire analysis and the application of various packages were conducted within the R programming environment. AUC calculations were carried out using the pROC package. For model construction, the glm package was used for logistic regression, and the rpart package was applied to develop tree-CART algorithms.

### 2.7. Statistical Analysis

The methylation levels of the urinary genes, as determined by qMSP, were computed as log2-transformed ΔCt ratio and represented as boxplots. The Ct ratios of the urinary methylation markers were employed to generate a receiver operating characteristic (ROC) curve and determine the area under the curve (AUC). The comparative data are shown as standard deviation (SD) with the significance between the means calculated using a two-tailed Student’s *t*-test. A *p*-value less than 0.05 was considered statistically different. All analyses and statistical graphs were depicted by using SPSS Statistics for Windows, Version 20.0 (IBM Corp., Armonk, NY, USA) and GraphPad Prism 10 (GraphPad Software, San Diego, CA, USA).

## 3. Results

### 3.1. Identification of BCa-Associated Methylation Sites from TCGA Bladder Cancer Dataset

The flowchart (Figure 1) depicts the developmental stages of identifying and validating the three-gene DNA methylation biomarkers for detecting BCa. In the discovery stage, we analyzed the DNA methylation profiles of ZNF671, OTX1, and IRF8 genes in TCGA BCLA through the UCSC Xena website. This graphics-based interactive platform provides access to the TCGA database and helps us identify potential hypermethylation sites closely associated with BCa.

This dataset provides access to DNA methylation profiles by DNA methylation array assay, transcriptomic profiles by RNA sequencing (RNA-Seq) analysis, and clinicopathologic phenotypes of the BCa tissues, which allowed us to specifically search for BCa-linked hypermethylated CpG sites that potentially serve as epigenetic biomarkers for BCa. In the current study, three hypermethylated CpG regions of ZNF671, OTX1, and IRF8 genes were selected. These methylation sites exhibited higher levels of methylation in BCa tumors than in non-BCa tissues (Figure 2a). In addition, we also analyzed the RNA-Seq data to compare the mRNA expression levels of the three genes between BCa and non-BCa samples. Our analysis shows that both ZNF671 and IRF8 exhibited significantly lower gene expressions in BCa tumors than in non-BCa tissues (Figure 2b), indicating potential tumor suppressive roles of the genes [44,45]. For the OTX1 gene, in contrast, it was observed that its expression was elevated in BCa, indicating a positive regulatory role in tumorigenesis [46]. Additionally, the survival curve analysis also reveals that increased OTX1 expression is linked to a poorer survival rate. Nevertheless, the expression levels of the other two genes (ZNF671 and IRF8) do not appear to have an impact on survival rates in this cohort, implying that epigenetic changes (herein DNA hypermethylation) may have a more significant effect on BCa status (Appendix A). In sum, we show that the elevated BCa-associated methylation sites identified from the TCGA dataset, even though they were from local tumor tissue analysis, can offer valuable cues for developing a urine-based DNA methylation diagnostic assay to detect BCa.

### 3.2. Clinical Evaluation of Elevated DNA Methylations in the BCa Urine Samples

To evaluate the feasibility of using urine DNA hypermethylation status of ZNF671, OTX1, and IRF8 genes for differentiating between BCa and non-BCa patients, we collected a total of 114 urine samples, consisting of 61 from BCa patients, 53 from patients non-BCa patients with mild lithangiuria, and 5 from healthy volunteers (Table 1). The urinary DNAs were purified, and further bisulfite conversions were performed. The levels of DNA methylation were measured by real-time quantitative methylation-specific PCR (qMSP) analysis. Our results show that all three methylation regions we assayed exhibit significant increases in DNA methylation in BCa urine samples compared to the non-BCa and healthy controls (Figure 3a), indicating these hypermethylation sites are BCa-specific. We further explored the potential application of these DNA hypermethylation markers for early-stage BCa detection. We sub-grouped the patients according to their disease stages into the non-BCa group, early-stage (Ta/T1), and late-stage (T2/T3). We found that all three gene hypermethylation biomarkers proved to be efficient in discerning early-stage BCa patients from those without BCa, suggesting a potential application of these markers for the prompt identification of BCa (Figure 3b).

### 3.3. Construction of Urine-Based DNA Methylation Diagnostic Models for Predicting BCa

To develop a robust diagnostic model for predicting BCa using urine DNA methylation assay, we further exploited the 114 urine samples for ROC analysis and sensitivity and specificity assessments. Eighty percent of these (total number = 90), comprising 50 BCa and 40 non-BCa samples, were randomly selected as the training set to construct the predictive models. The remaining 20% (n = 24), consisting of 12 BCa and 11 non-BCa samples, comprised the testing set for assessing new data. In both the training and testing sets, the distribution of classes mirrored the composition of the original dataset. The performance metrics of the logistic regression and decision tree models on both the training and testing datasets are demonstrated in Table 2 and Table 3, respectively. The models utilizing a single biomarker, representing ZNF671, OTX1, and IRF8, achieved training set accuracies of 84.62%, 79.12%, and 74.73% (Table 2) and testing accuracies of 73.91%, 73.91%, and 69.57% (Table 3), correspondingly. In addition, the individual biomarkers attained AUC values of 0.8649, 0.8214, and 0.8071, respectively (Figure 4a). Next, we found that the combined use of all three biomarkers generally outperformed the performance of single-marker models (Figure 4b). For instance, logistic regression boosted its training and testing accuracy to 0.8681 and 0.7826, respectively, with an AUC increase of 0.8950 (Figure 4b). The CART decision tree model outperformed logistic regression, achieving training and testing accuracies of 87.91% and 82.61%, respectively, and an AUC of 0.9077 (Figure 4b). Interestingly, although we initially included all three biomarkers in model building, the decision tree model autonomously selected only two, ZNF671 and OTX1, yet demonstrated excellent performance (Figure 5). The tree structure of the model indicates that the ZNF671 hypermethylation marker alone is capable of distinguishing between BCa and non-BCa in urine samples (ΔCt Cutoff = 10.4). Samples below this cutoff (i.e., hypermethylation status) suggest a 92.68% likelihood of indicating BCa. Furthermore, for the samples with ΔCt values equal to or greater than 10.4, OTX1 levels are considered. The ΔCt of OTX1 level below 8.23 suggests about a 63.64% chance of BCa, whereas levels equal to or greater than the Ct value nearly confirm non-BCa with about a 90% probability.

In sum, the promising potential of hypermethylated biomarkers ZNF671, OTX1, and IRF8 in predicting BCa is evident in this study. Furthermore, combining these markers could improve predictive accuracy. Notably, the decision tree model using only ZNF671 and OTX1 demonstrates satisfactory performance with specific cutoffs and criteria, providing a feasible approach for predicting BCa and tracking disease progression through the urine methylation assay.

## 4. Discussion

In the current study, we investigated whether the methylation status of three genes, ZNF671, OTX1, and IRF8, could serve as potential biomarkers for detecting BCa. For this purpose, we initially analyzed the dataset from the TCGA bladder cancer (BLCA) cohort, which consists of 411 BCa samples and 17 normal tissues [38]. Utilizing this dataset that provides information on RNA-sequencing, DNA methylation array, and various clinical phenotypes, we employed the existing database as a robust platform for identifying potential biomarkers during the discovery phase. Previous studies, including our own [47], have revealed that ZNF671 exhibits tumor suppressor properties [44,45,48]. Analysis of the TCGA BCLA cohort also consistently demonstrated a lower level of mRNA expression in BCa patients compared to non-BCa or normal tissues. This suggests the diminished expression of ZNF671 could be attributed to epigenetic suppression caused by DNA hypermethylation within its promoter region. Hence, this inverse relationship between DNA hypermethylation and clinical outcomes, including mRNA expression, offers a potential window for identifying cancer-specific DNA hypermethylation sites. In contrast, orthodenticle homobox1 (OTX1) acts as a transcription factor and has been shown to function as a potential oncogene, with its overexpression observed in various cancers, including breast cancer, colorectal cancer, hepatocellular carcinoma, and bladder cancer [46,49,50,51,52]. In addition to the promoter region, it could be advantageous to investigate hypermethylation sites in the 3′-untranslated region (3′-UTR) as potential hypermethylation biomarkers for cancers. A recent study showed that a hypermethylation site located in the 3′UTR of OTX1 has been utilized for detecting bladder cancer from urine sediments [33]. Our TCGA BLCA studies also show additional hypermethylation sites in OTX1 3′-UTR that could be useful for developing specific hypermethylation biomarkers for BCa.

Our urine DNA methylation diagnostic panel encompasses a trio of genes that includes interferon regulatory factor 8 (IRF8). IRF8 is a transcription factor known for its role in regulating type I interferon (IFN-I)-dependent signaling pathways [53,54,55]. IRF8 also plays a role in the development of myeloid cells [56]. Studies on mice with a knockout of the Irf8 gene revealed that the absence of CD8α+ dendritic cells and macrophage functions, along with the production of abnormal granulocytes, was linked to the development of an atypical form of chronic myelogenous leukemia (CML) [57,58]. Of note, several studies have shown that IRF8 is involved in inhibiting multiple non-hematopoietic cancer cell growth and progression, indicating its potential tumor-suppressive activities [59,60] and its expression is regulated by DNA methylation [61,62]. BCa is a heterogeneous disease that can be classified into non-muscle-invasive bladder cancer (NMIBC) and muscle-invasive bladder cancer (MIBC), with different clinical outcomes and management strategies. The current protocol for diagnosing and surveillance of the disease is cystoscopy, which is an invasive and costly procedure that can cause discomfort and complications for patients [22]. However, the other standard method is urine cytology, which is the microscopic examination of urine cells and is often used as an adjunct to cystoscopy, but it has low sensitivity for low-grade tumors and high interobserver variability [20]. Therefore, there is a need for reliable and non-invasive urine biomarkers that can improve the detection and monitoring of bladder cancer, especially NMIBC, which has a high recurrence rate and requires frequent surveillance. Hence, our urine-based three-gene DNA methylation diagnostic panel shows potential as a valuable tool for frequent monitoring of NMIBC. While showing potential, the diagnostic panel still requires further validation in large-scale prospective studies before it can be implemented in routine clinical practice.

The use of liquid biopsy has shown great potential as a convenient and non-invasive method for diagnosing cancers [63]. One of the promising approaches is to detect abnormal levels of DNA methylation from body fluids including blood or urine samples [64,65]. In this study, we developed a urine-based non-invasive diagnostic panel for BCa detection depending on three-gene DNA methylation measurements of urine sediments that contain exfoliating tumor cells. Following performance evaluation using ROC analysis with two different machine learning algorithms (logistic regression and decision tree), although minor variances were observed between the two, we determined that the decision tree algorithm was utilized as the primary method for constructing the diagnostic model. Notably, the hypermethylated ZNF671 marker holds considerable promise in distinguishing between bladder cancer and non-BCa urine samples by the tree structure, achieving a 92.7% probability (Figure 4b). In addition, the use of the second hypermethylation marker, the OTX1 gene, in the tree structure resulted in a better predictive performance. Corroborating this result, several studies also have utilized DNA methylation of OTX1 as a biomarker for detecting BCa. For example, several studies found that combining DNA methylation analysis using OTX1, ONECUT2, and TWIST1 genes with mutation analysis using FGFR3, TERT, and HRAS provided an accurate prediction model for selecting patients with hematuria for cystoscopy [37,66]. Recently, Chen et al. (2020) developed a urine-based diagnostic model for BCa using OTX1 and SOX1-OT1 methylation markers. Their model demonstrates high sensitivity and specificity for detecting early-stage BCa, minimal residual disease, and recurrence BCa [33], suggesting OTX1 is a promising biomarker for BCa.

While the three-gene DNA methylation diagnostic panel showed a high promise in detecting early-stage BCa, there were some limitations to be considered in this study. Firstly, while MS-qPCR offers a highly sensitive technique for quantifying DNA methylation, those urine samples from BCa patients with Ct values above the cutoff (e.g., Ct ≥ 38) were categorized as having low or non-DNA methylation. This could be attributed to an extremely low number of exfoliating tumor cells in the urine rather than accurately reflecting DNA methylation levels. To enhance the detection limits in this study, a new technology with higher sensitivity, like digital PCR, could be employed to improve the sensitivity [67,68]. Secondly, since the analysis of specific DNA methylation sites in urine sediments allowed us to distinguish between BCa and non-BCa cases, the selected methylation sites used in this study were based on local tumor DNA methylation profiles from the TCGA BCLA database. However, it will be important to consider that the mechanisms of phenotypic plasticity resembling aspects of epithelial–mesenchymal transition (EMT) and anoikis resistance, which is mediated by epigenetic alterations, play a crucial role in transforming localized tumor cells into circulating tumor cells (CTCs) during cancer progression [69,70]. CTCs are cancer cells that detach from the primary tumor and enter the circulation, where they can potentially spread to other organs. CTCs have been regarded as diagnostic and prognostic biomarkers for various cancers, including bladder cancer [71]. Several methods have been developed to isolate and identify CTCs in urine samples, such as EpCAM-based immunomagnetic separation [72]. Recent studies have shown that CTCs carrying specific tumor biomarkers were associated with worse overall survival (OS) and recurrence-free survival (RFS) in NMIBC patients who underwent transurethral resection of bladder tumor (TURBT) [73]. Based on this notion, leveraging single-cell epigenome–transcriptome profiling of the CTCs presenting in urine or blood samples could offer a more precise identification of cancer-specific biomarkers present in exfoliating tumor cells.

## 5. Conclusions

In this study, we demonstrated a urine-based DNA methylation diagnostic panel based on the three hypermethylation loci of ZNF671, OTX1, and IRF8 for detecting and predicting BCa. This diagnostic panel achieved an accuracy of 82.6%, sensitivity of 75%, and specificity of 90.9% in distinguishing between BCa and non-BCa by urine samples. Additionally, the diagnostic panel is also suitable for identifying early-stage BCa, such as NMIBC, which has a high recurrence rate and requires frequent surveillance. While showing promise as a surveillance tool for BCa, further study is necessary to validate the clinical effectiveness of BCa detection and understand the biological relationship between hypermethylation loci of ZNF671, OTX1, and IRF8 and BCa tumorigenesis.

## Figures and Tables

**Figure 1 diagnostics-14-00468-f001:**
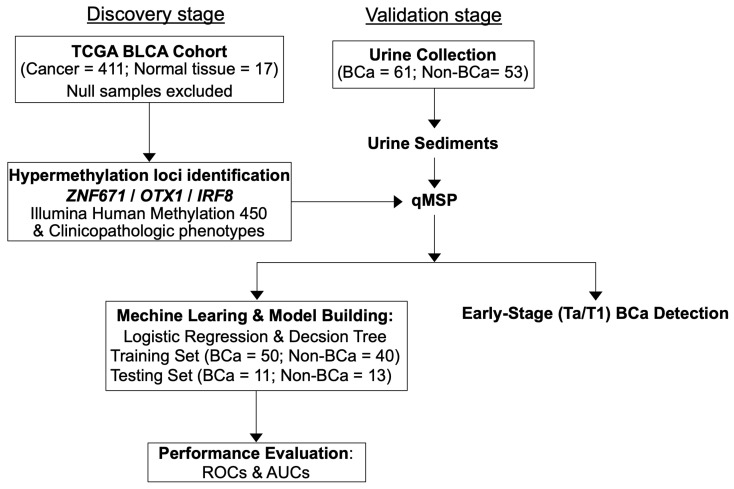
Flowchart for the development stage of BCa-specific methylation diagnostic panel in this study.

**Figure 2 diagnostics-14-00468-f002:**
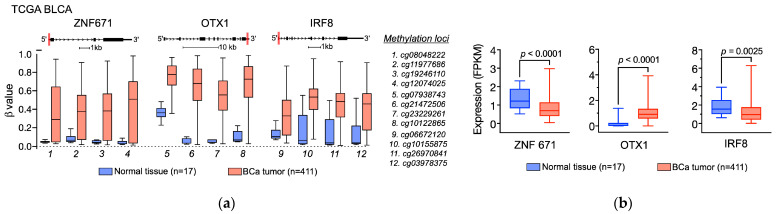
Identification of BCa-specific hypermethylation sites on ZNF671, OTX1, and IRF8 genes using the TCGA BLCA dataset. (**a**) Comparison of methylation levels measured by the Human Illumina Methylation 450 Array method in normal (blue boxes) and BCa (red boxes) tissues. The gene structures and the methylation sites (indicated as red lines) of each gene are indicated above the box plot. The methylation loci are indicated beside the box plot. (**b**) Gene expression levels were measured by RNA sequencing in normal (blue boxes) and BCa (red boxes) tissues. The *p* values were calculated by the Student’s t-test.

**Figure 3 diagnostics-14-00468-f003:**
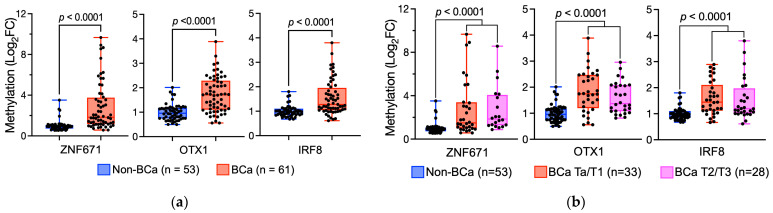
Clinical validation of the hypermethylated regions on ZNF671, OTX1, and IRF8 genes as BCa-specific biomarkers from urine samples. (**a**) Methylation levels of the ZNF671, OTX1, and IRF8 genes were measured by real-time quantitative methylation-specific PCR in urine samples. The methylation levels are represented by using box plots with log_2_ fold change (FC). (**b**) Comparison of sub-grouping BCa-stages (i.e., Non-BCa, Ta/T1, or T2/T3 are indicated) of hypermethylated ZNF671, OTX1, and IRF8 genes of urine samples. Data are represented by box plots. *p* values were calculated by the Student’s *t*-test.

**Figure 4 diagnostics-14-00468-f004:**
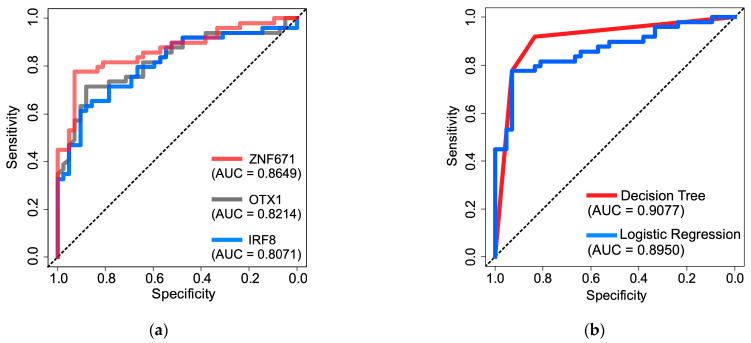
Performance assessment of diagnostic panel constructed using machine learning algorithms. (**a**) Receiver operating cure (ROC) analysis of individual hypermethylated gene as indicated. Area under the curve (AUC) values are shown. (**b**) ROC analysis for a combination of ZNF671, OTX1, and IRF8 with logistics regression and decision tree as indicated. AUC values are shown.

**Figure 5 diagnostics-14-00468-f005:**
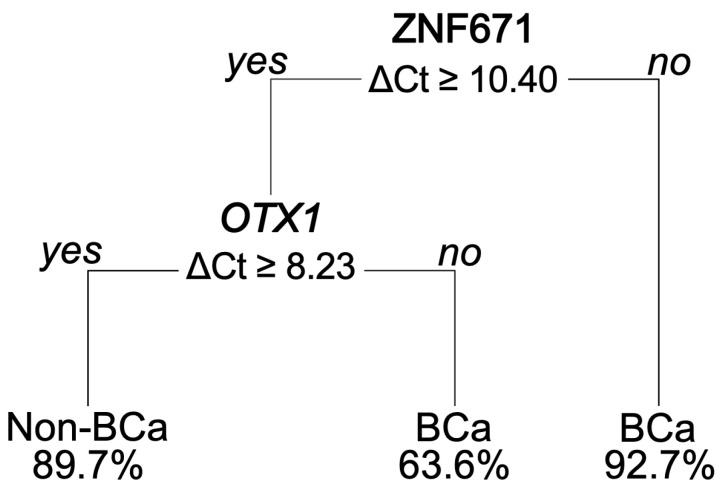
A tree structure derived from the decision tree algorithm. A lower Ct value indicates a higher methylation level.

**Table 1 diagnostics-14-00468-t001:** Clinicopathologic and demographic features of subjects in this study.

Characteristics	Training Set		Testing Set	
	BCa ^1^ (*n* = 50)	Non-BCa (*n* = 40)	BCa (*n* = 11)	Non-BCa (*n* = 13)
Sex				
Male	29	27	7	4
Female	21	13	4	9
Age (year)	69.1 ± 9.9	61.9 ± 12.5	61.7 ± 5.6	53.0 ± 15.6
Stage				
Ta	12 (24.0%)		2 (18.2%)	
T1	15 (30.0%)		4 (36.4%)	
T2	9 (18.0%)		1 (9.1%)	
T3	11 (22.0%)		4 (36.4%)	
T4	3 (6.0%)		0 (0%)	
Grade				
Low	7 (14.0%)		2 (18.2%)	
High	41 (82.0%)		9 (81.8%)	
Unknown	2 (4.0%)		0 (0%)	

^1^ BCa: Bladder cancer.

**Table 2 diagnostics-14-00468-t002:** Performance analysis of training set.

Parameters (%)	Marker Gene			Model	
	ZNF671	OTX1	IRF8	Logistic Regression ^1^	Decision Tree ^1^
Accuracy	84.62	79.12	74.73	86.81	87.91
Sensitivity	77.55	71.43	61.22	85.71	91.84
Specificity	92.86	88.10	90.48	88.10	83.33
Precision	92.68	87.50	88.24	89.36	86.54
F1 Score	84.44	78.65	72.29	87.50	89.11
AUC ^2^	86.49	82.14	80.71	89.50	90.77

^1^ Model construction considering ZNF671, IRF8, OTX1 as candidate markers; ^2^ AUC: area under curve.

**Table 3 diagnostics-14-00468-t003:** Performance analysis of testing set.

Parameters (%)	Marker Gene			Model	
	ZNF671	OTX1	IRF8	Logistic Regression ^1^	Decision Tree ^1^
Accuracy	73.91	79.12	69.57	78.26	82.61
Sensitivity	77.55	71.43	50.00	66.67	75.00
Specificity	92.86	88.10	90.71	90.91	90.91
Precision	92.68	87.50	85.71	88.89	90.00
F1 Score	84.44	78.65	63.16	76.19	81.82

^1^ Model construction considering ZNF671, IRF8, OTX1 as candidate markers.

## Data Availability

Data are available by contact with the corresponding author.

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
