# Peer review of "Hypermethylation Loci of ZNF671, IRF8, and OTX1 as Potential Urine-Based Predictive Biomarkers for Bladder Cancer"

_diagnostics, 2024, doi:10.3390/diagnostics14050468_

Round 1

Reviewer 1 Report

Comments and Suggestions for Authors

It is an novel experimental idea to check gene DNA methylation for prediction of bladder cancer development. The authors found some significant gene DNA hypermethylation of bladder tumor tissue from TCGA database. Following that, the authors collected urine samples from clinical patients to prove the experimental hypothesis. Finally, the authors concluded that hypermethylation of ZNF671OTX1 and IRF8 could be a prediction panel. But some questions were needed to clarify :

1.      In the abstract, the author developed gene panel(ZNF671OTX1 and IRF8) to detect bladder cancer. Besides, the author used the panel for prognosis of bladder cancer. The manuscript did not provide the survival results. Please explained it or provided the supplement results.  

2.      In material and method, the enrolled time … “ All participants were enrolled in our case-control retrospective study from November 2021 to November 2021”. It was needed to be corrected.

3.      The authors analyzed the bladder tumor tissue from TCGA dataset and point out the specific hypermethylation of gene(ZNF671OTX1 and IRF8) to bladder cancer. But the authors used the urine sample to detect DNA methylation. The confounding factors in urine sample were difficult to resolve. How to explain the ZNF671OTX1 and IRF8 in urine specific to bladder tumor and not to other cell. 

Comments on the Quality of English Language

well

Reviewer 2 Report

Comments and Suggestions for Authors

This study aims to develop a urine DNA methylation diagnostic panel for distinguishing between BCa and non-BCa. I believe that the study has sufficient merit to be considered for publication on diagnostics, although major revisions are required.

MAJOR COMMENTS

-       Introduction: Provide a more detailed review of the limitations of current diagnostic methods. I suggest discussing in this first section the risk factors associated with bladder cancer, including the recent research on new biomarkers (doi: https://doi.org/10.1016/j.euros.2023.11.003) (doi: 10.3390/ijms241310846).

Clearly state the objectives of the study. Emphasize the clinical implications of early bladder cancer diagnosis.

-       Materials and Methods: Provide additional information on procedures ensuring the accuracy of DNA methylation measurements. Justify the specific choice of machine learning techniques more thoroughly.

-       Discussion: Expand on the implications of results for clinical practice and future research.

Discuss limitations more thoroughly. Provide recommendations for future research.

-       Conclusion: Effectively summarizes main results and implications.

Include a brief statement on practical implications and future research directions, along with a synthesis of main study limitations.

Comments on the Quality of English Language

Moderate editing of English language required

Round 2

Reviewer 2 Report

Comments and Suggestions for Authors

I believe that the study has sufficient merit to be considered for publication